# Nano Edible Coatings and Films Combined with Zinc Oxide and Pomegranate Peel Active Phenol Compounds Has Been to Extend the Shelf Life of Minimally Processed Pomegranates

**DOI:** 10.3390/ma16041569

**Published:** 2023-02-13

**Authors:** Hosam Aboul Anean, L. O. Mallasiy, Dina M. D. Bader, Heba A. Shaat

**Affiliations:** 1Food Engineering and Packaging Department, Food Technology Research Institute (FTRI), Agriculture Research Center (ARC), Giza 12619, Egypt; 2Department of Home Economics, Faculty of Science and Arts in Tihama, King Khalid University, Muhayil Asir 61913, Saudi Arabia; 3Chemistry Department, Muhayil College of Science and Arts, King Khalid University, Muhayil Asir 61913, Saudi Arabia; 4Food Science and Technology Department, Faculty of Home Economic, Al-Azhar University, Tanta 31732, Egypt

**Keywords:** chitosan, ZnONP, pomegranate, mechanical, viscosity, vapor, zeta particle size, color, light transmittance, scanning electron microscopy and edible coating film

## Abstract

Edible coating and film from chitosan and incorporating it with the action of ZnONPs on active phenol compounds from extracts of pomegranate peel (PPE) The physical and chemical properties of edible films composed of zinc oxide ZnONPs and active phenol compounds extracted from pomegranate peel (PPE) were investigated. Adding ZnONPs with active phenol compounds from extracted pomegranate peel(PPE) to chitosan films can provide safe edible films, decrease microbial growth and consequently prolong the shelf life of pomegranates, as well as improve the physiochemical stability of the pomegranate. The substances used in this experiment were film with a (A) extract of pomegranate peels (PPE), 5% (0.1%), (B)ZnONPs 1% (0.02%), (C) ZnONPs 2% (0.04%), (D) ZnONPs 3% (0.06%), (E) ZnONPs 1%/PPE1% (0.02%), (F) ZnONPs 2%/PPE2% (0.04%), (G) ZnONPs 3%/PPE3% (0.06%) wt% of chitosan on quality attributes and prolonging the shelf life of pomegranates were stored in plastic containers at 2 °C and 90–95% relative humidity for 20 days. The treatments of (G) ZnONPs 3%/PPE3% (0.06%) loaded on chitosan as well as chitosan and (D) ZnONPs 3% (0.06%) reduced the weight loss, had excellent microbial count until 20 days of storage, and recorded the lowest microbial count and mould & yeast colonies. Other chemical properties, such as total soluble solids content, acidity, anthocyanin content, firmness, and ascorbic acid, were investigated. Results indicated that ZnONPs 3%/PPE3% (0.06%) loaded on chitosan or ZnONPs 3% (0.06%) are the best treatments for preserving pomegranate arils. It was found that the best measurements were that the film-forming nan emulsion solutions decreased by E% 110 nm and B% 134 nm. Nano followed treatment, F% 188.7 nm, compared to nano edible films, which were A 0% 1312 nm.

## 1. Introduction

Polysaccharide-based edible coatings and their de-privatives are widely used to fabricate edible coatings and films to enhance food products’ shelf life and quality retention. They are proven to possess good oxygen barrier properties but are hydrophilic; edible coatings derived from polysaccharides lack good moisture barrier properties. Polysaccharide-based edible coatings are colourless and have lower caloric content. In addition, it can be used to extend the shelf life of food products such as fruits, vegetables, and meat [1]. 

Chitosan is a partially deacetylated polymer of acetyl glucosamine that is a naturally occurring linear cationic polysaccharide. Chitosan is obtained through alkaline deacetylation of chitin, which can be extracted from various crustacean shells, fungal cell walls, and other biological materials. The antimicrobial activity of chitosan depends on its concentration, molecular weight, and deacetylation degree. Chitosan solutions in various organic acids can be prepared, which form flexible, transparent, and rigid films that are suitable oxygen barriers upon drying. In food industries, chitosan has been extensively used for direct surface coating of meat and fruit products to reduce food deterioration and water loss, as well as delay the ripening of fruits. In 2001, chitosan was classified as safe by the US FDA and was thus considered safe to be used as a food preservative [2]. Advances in the preparation of nan systems that incorporate ingredients acceptable for food products have made it possible to explore the functional modifications of edible coatings that integrate nan emulsions, polymeric nanoparticles, nanofibers, solid lipid nanoparticles, nanostructured lipid carriers, nanotubes, nan crystals, nan fibers, or mixtures of organic and inorganic nan-sized components. Generally, these nan systems are incorporated into polysaccharides or protein matrices called “nano composites,” which are defined as the combination of two or more materials to form a mixture that improves the properties of a component in which at least one of them has a nan metric scale [3]. The development of nanocomposites has allowed edible coatings to be used as temporal distribution systems” that release active substances from a material film to the food to improve conservation [4]. A previous study mentioned that nano-scale zinc oxide notably induced partial toxicity through releasing ROS radicals in plant and animal cells. However, the cells can be grown in such conditions without [5]. Because of their ability to survive in harsh environments, zinc particles, particularly zinc oxide (ZnO), are being widely proposed to be used as antimicrobial agents with a broad range of other applications due to their broad spectrum of other applications. The antimicrobial activity of zinc oxide (ZnO) particles was proposed due to the emission of zinc ions (Zn^2+^), which are able to penetrate into the bacteria’s cell wall and affect the cytoplasmic content in the cell that leads to the death of bacteria. The incorporation of zinc oxide nanoparticles into gelatin was observed by [6]. which revealed that the film showed a higher inhibitory effect against gram-negative bacteria (*Pseudomonas aeruginosa*) than gram-positive (*Enterococcus faecalis*) bacteria. The findings supported the hypothesis that ZnO induced a photo catalytic mechanism due to its semi-conductive properties, resulting in the formation of reactive oxygen species (ROS) and H2O2, which damage the cell wall structure of bacteria [7,8].

The toxic effects of ZnO NPs are due to their solubility, resulting in increased intracellular [Zn^2+^]. This results in cytotoxicity, oxidative stress, and mitochondrial dysfunction. It is unclear whether this is due to NP uptake by cells or NP dissolution in the medium. In vivo airway exposure poses an important hazard. Inhalation or instillation of the NPs results in lung inflammation and systemic toxicity. Reactive oxygen species (ROS) generation likely plays an important role in the inflammatory response. The NPs do not, or only to a minimal extent, cross the skin; this also holds for sunburned skin [9]. The antimicrobial effect, UV-blocking property, and capability to reinforce the nanocomposites have resulted in the growing application of ZnO as a multifunctional component of packaging films for food applications. It has been widely used in solar cells, semiconductor diodes, sensors, ceramics, UV light-emitting devices, visitors, catalysts, etc. Furthermore, a new application of ZnO was identified after discovering its antimicrobial activities in 1995. In recent years, several scientific studies have been conducted on developing composite films by combining different biopolymers with nano-ZnO [10,11]. The principal interests in the food packaging field are developing biodegradable packaging materials that can replace conventional petroleum-based food packaging materials and developing functional packaging materials. In addition to the traditional food packaging functions of containment and protection, multifunctional food packaging materials play an essential role in prolonging the shelf life of food by maintaining food quality and preventing food degradation and microbial contamination [12]. The pomegranate (*Punica granatum* L.) belongs to the Punicacea family. It is one of the most important and commercial horticultural fruits that is generally very well adapted to the Mediterranean climate. It has been cultivated extensively in Iran, India, and some parts of the USA, China, Japan, and Russia. Pomegranate fruits are consumed fresh or processed as juices, jellies, and syrup for industrial production. Different parts of its tree (leaves, fruits, and bark skin) have traditionally been used for their medicinal properties and other purposes such as tanning [13]. minimal processing Operations alter fruits’ integrity, bringing about adverse effects on product quality such as browning, off-flavor development, and texture breakdown. In addition, the presence of microorganisms on the fruit surface may compromise the safety of fresh-cut fruit. The search for methods to retard these adverse effects is of great interest to all the stakeholders involved in producing and distributing fresh fruits. Edible coatings may extend the shelf-life of fresh-cut fruits by reducing moisture and solute migration, gas exchange, respiration and oxidative reaction rates, and by reducing or suppressing physiological disorders [14]. Edible coatings have a high potential to carry active ingredients such as anti-browning agents, colorants, flavors, nutrients, spices, and antimicrobial compounds that can extend product shelf life and reduce the risk of pathogen growth on food surfaces. These studies on fresh fruits are somewhat limited, and their industrial implementation is still developing. In this sense, the main goal of this article is to review and update the information available on the use of edible coatings as carriers of food ingredient antimicrobials to improve the safety, quality, and functionality of fresh fruits [15]. This work aims to produce edible films from zinc oxide and active phenol compounds of PPE nanoparticles loaded on chitosan films. The physical and mechanical properties, including viscosity, vapor, zeta particle size emulsion, color, light transmittance, and SEM, were studied to select the best edible coating for improving pomegranate fruit storability under cold storage. as well as minimizing decay and microbial growth during the storage period.

## 2. Preparation of Different Materials and Methods

### 2.1. Materials

‘Manfalouty’ pomegranate (*Punica granatum* L.) fruits were obtained from a private orchard in Assiut Governorate. This study was conducted in 2021 and 2022. were harvested in mid-October, fully matured as followed in commercial practice, and transported to the postharvest handling lab. at the Horticulture Research Institute (Giza Governorate, Egypt) to study the effect of different postharvest treatments on quality and pomegranate arils. The substances used in this experiment were: chitosan from Acros-organics Company, Morris Plains, NJ, USA. Citric acid, sodium citrate, and sodium hydroxide were obtained from (El Nasr Pharmaceutical Chemicals Company, Cairo, Egypt). Glycerin and calcium hypochlorite were obtained from (El-Gomhouria chemical company, Cairo, Egypt). ZnONPs were obtained from the Nanotechnology & Advanced Material Center Lab, Agriculture Research Center, Giza, Egypt. Ethyl alcohol is 95% produced by El-Gamhoria Company Egypt. Tween was obtained from Win Lab Company (London, UK). Glacial acetic acid was obtained from (Acmatic For Chemicals & Lab. Equipment Company, Cairo, Egypt).

### 2.2. Treatments

#### 2.2.1. Storage Treatments of Studied Pomegranate Fruits

The dipping of fresh pomegranates was peeled and the arils were separated from the tissues. Then, the arils were washed and treated with a disinfecting solution of calcium hypochlorite (0.25 g/L distilled water) for one minute before being air-dried. The arils are created with a disinfecting solution of calcium hypochlorite (0.25 g/L distilled water) for one minute before being air-dried. The arils were treated as follows: Fresh fruits were (A) extracted pomegranate peels (PPE) 5% (0.1%), (B)ZnONPs 1% (0.02%), (C) ZnONPs 2% (0.04%), (D) ZnONPs 3% (0.06%), (E) ZnONPs 1%/PPE1% (0.02%), (F) ZnONPs 2%/PPE2% (0.04%), (G) ZnONPs 3%/PPE3% (0.06%) Chitosan content in packaged coated fruitsPomegranate peeled cubes were drained after dipping and packaged in plastic trays with approximately 250 g. After that, all boxes were cold stored at 2 °C and 90–95% RH for 20 days and kept in carton boxes. All samples were kept after packaging. Samples were taken at weekly intervals to determine the physical and chemical changes during the storage period. Pomegranate samples were withheld for analysis on a regular basis.

Prepared crude phenolic compound extraction from pomegranate peels(PPE):

The air-dried ground (80 mesh) plant material (20 g for each sample) was extracted with each of the solvents ethanol (ethanol: water, 80:20 *v*/*v*) and aqueous methanol (methanol: water, 80:20 *v*/*v*) (200 mL) for 6 h at room temperature in an orbital shaker in a water bath in separate experiments. The extracts were separated from the residues by filtering through Whitman No.1 filter paper. The sediments were extracted twice with the same fresh solvent, and the combined extracts were used. Using a rotary evaporator, the combined extracts were concentrated and freed of solvent under reduced pressure at 45 °C. According to [16]. the dried crude concentrated extracts were weighed to calculate the yield and stored in a refrigerator (−4 °C).

#### 2.2.2. Preparation of the Film

The edible coatings and film solutions of chitosan (2 g/2 mL glacial acetic acid) were prepared separately by dissolving each in 100 mL of deionized water while stirring with a magnetic stirrer at room temperature. Subsequently, the required amount of ZnONPs (1 to 3 wt% of Chitosan) was dispersed in 100 mL of distilled water and added using an ultrasonic bath for 30 min at different chitosan solutions. Then, add a suspension of ZnONPs to it and mix it in a homogenizer. The mixed in the ratios were divided into seven groups: (A) 5% (0.1%) pomegranate peel extract (PPE), (B) 1% (0.02%), (C) 2% (0.04%), (D) 3% (0.06%), (E) 1%/PPE1% (0.02%), (F) 2%/PPE2% (0.04%), (G) 3% (0.06%), (H) 3% (0.06%), (G) 3% (0.06%), (H) 3% (0.06%),The composition of the solution mixture was modified. The film-forming solution described above was modified by adding 5% (0.1%) of active phenol compounds from pomegranate peel extract (PPE). To stabilise the emulsion of active phenol compounds from pomegranate peel extracts (PPE), Tween-20 was also added to the chitosan solution with a ratio of 0.2% of the active phenol compounds from pomegranate peel extracts (PPE). Then, it is added to the solution and left to dissolve for 30 min until a clear solution is obtained. The film’s seven nano-edible coatings and films were combined with ZnONPs and ZnONPs/PPE by a suspension solution consisting of 1% (0.02%), 2% (0.04%), and 3% (0.06%) wt% of chitosan in 10 mL of alcohol (95%), plus 100 mL of deionized distilled water with the incorporation to contain 1.2% *v*/*v* glycerol. After adjusting the pH to 5.6 with 1 M NaOH and homogenizing the solution, it was heated in a magnetic stirrer at 75 °C for 30 min. The edible film-forming nanoedible coating and films combined with [17].

Table 1 show up the formulas for nano edible coating & films combined with ZnONPs and ZnONPs/PPE. 

**1.** The parameters (shear rate and shear stress) to choose the best solutions for edible film nano edible coating and films solutions combined with ZnONPs and ZnONPs/EPP were measured using Brookfield Engineering Labs. DVN EXT. BROOKFIELD AMETK- At room temperature, the most appropriate solutions are selected. The methodology of work Put the sample transformer in a water bath at a constant temperature to maintain the desired temperature and then run the viscosity device between 10 and 60 rpm The spindle sc4-18 was selected for measurement.


**2. Determination of nanoparticle zeta size**


The edible coating solutions were measured on the following nano-devices: Malvern Zeta Sizer nano series (Nano ZS), UK. The wavelength range is 0.6–6000 nm, and the zeta range (mV) is (−200:200 mV).

**3.** Measurement of edible film nano-edible coating & film solution combined with ZnONPs and ZnONPs/PPE using microscopy electron scanning electron by:

Inspect S 150A SPUTTER COATER SEM Schematic Overview-Quanta FEG250 with field emission gun, FEl company Netherlands, Machine type inspect [18].

**4. Film thickness:** The film thickness was measured for the preparation of nano edible coating and films solution combined with ZnONPs and ZnONPs/PPE to produce edible film at different treatments of A 0%, B 0.02%, C 0.04%, D 0.06%, E 0.02%, F 0.04%, and G 0.06%, measured with a digital micrometre (Mitutoyo type Digital Indicators, the company’s models: pk-1012 E, Japan) where the film strip was placed between the jaws of the micrometre and the gap and then slowly closed and took an average of three readings [19].

**5. Color:** Inside colour measurements The nano-edible coating & film solution combined with ZnONPs and ZnONPs/PPE values were measured with a Colorimeter Minolta chroma meter. “CR-200” is the Cyril number. The calibration test was carried out on a plank whiteboard before use, and then the colour changes were measured in the treatments for the value indicating luminance and the value indicating the yellow colour area [20]. and measurement of colour using a scale of portable colours based on the L*a*b* colour scheme The L* represents where the luminance value ranges from 0 (the darkest black) to 100 (the brightest white), and the a* represents “redness” or “greenness,” which ranges from +60 for the absolute red to -60 for the absolute green. The b* value is “yellow” or “blue”. Fans ranging from +60 to absolute blueFast results with a D65 light factor 8 mm aperture screen The Colorimeter is automatically calibrated in black and white Placing a video sample on a standard surface The colour difference can be calculated as follows:
ΔE* =√ (L* − L*s) ^2^+(a * − a*s) ^2^ + (b* − b*s)^2^
where L*, a*, and b* represent the film sample values, and L*s, a*s, and b*s represent the standard whiteboard values.As a background, Maria-Ioana Socaciu et al. (2020) represent values as background and take three readings for each film. and E = (L)^2^ + (a)^2^ + (b)^2^, where L, a, and b represent the colour difference between the film sample and the standard plate, respectively. According to [21,22].

**6. Light transmittance:** spectrum transmission meter measurement of the barrier and transparency of the films by measuring the light transmittance ratio at wavelengths T395, 430, and 550 nm, respectively, using a LINSHANG LS108 unit % BL Light Penetration by [17]. The light transmittance ratio and transparency value were determined using a dual-beam UV-VIS spectrophotometer (SPECTRUM VIS (patent): Serial No. 2012203020234 China) Made in the China by reading the absorbance of a film sample at wavelengths between 395, 430 and 550 nm. The transmittance is the percentage of incident light that passes through a film sample and is determined by the effective absorption and scattering of light by a transparent material having a transmittance above 90% [23]. Cut the film sample into 3 strips (0.7 cm × 3 cm) and place each film strip on a cuvette. Then read the absorbance of the sample against the blank cuvette. Then the value of the absorbance is converted to the transmittance ratio [24].

**7. % Solubility in water:** The nano-edible coating and film solutions were combined with ZnONPs and ZnONPs/PPE at different treatments. All A 0%, B 0.02%, C 0.04%, D 0.06%, E 0.02%, F 0.04%, and G 0.06% samples were first dried in a desiccator containing calcium chloride. Then the dry piece of 500 mg nano-film was immersed in beakers containing 50 mL of distilled water at room temperature for 24 h with gentle spin rocker incubation. Then the films were removed from the water and returned to the desiccator by weighing known and constant, then calculating weight loss in water as a percentage of water weight loss based on the dry film as follows: % weight loss = initial dry weight-final dry weight × 100/initial dry weight according to [16].


**8. Measuring the mechanical properties of a prepared nano edible coating and film solution that has been combined with ZnONPs and ZnONPs/PPE to produce edible coating and films.**


Measurement of edible film nano-edible coating and films solution combined with ZnONPs and ZnONPs/PPE at various treatments; measurement of tensile properties (tensile strength, elongation) using a texture analyzer of type CT3.The edible film nano biopolymer product is bacterial at different treatments A = 0%, B = 0.02%, C = 0.04%, D = 0.06%, E = 0.02%, F = 0.04%, G = 0.06%. It was cut into 3 × 5 cm strips. They were held At each jaw end, the jaws were then moved partly at the exact speed until the modulus of the youngsters was automatically recorded according to [19].

**9. Measuring water vapour permeability (VWP).** The ASTM E96-95 method was used to determine the water vapour transmission rate [g/s·m^2^]. and water vapour permeability through the nano-edible coating and film solution combined with ZnONPs and ZnONPs/PPE. A round test cup was used to determine the VWP of the membrane nanoparticles upon different processing: A 0%, B 0.02%, C 0.04%, D 0.06%, E 0.02%, F 0.04%, and G 0.06%. The membrane was first cut into a round shape larger than the inner diameter of the beaker. The beaker was filled with 50% distilled water, the membrane was sealed on top with paraffin oil, and then the beakers were placed in a desiccator containing calcium chloride. Hourly cup weights were recorded over 10 h, and samples from each film were tested. Linear regression was used to estimate the slope of this line in g/hr. The water vapour transfer rate (WVTR) and water vapour permeability were determined using the following:

The rate of water vapour transmission [g/(s·m^2^)] The water vapour permeability through films was determined gravimetrically using the ASTM method E96-95.
WVPR = Δm/ΔtA  WVP = WVPR·L/ΔRH
where, Δm/Δt is the moisture gain weight per time (g/S), A is the surface area of the film m2, L is the film thickness (mm) and ΔRH is the difference in relative humidity. (ASTM E96-95) [25].


**10. Measurmeant of gas permeability:**


Gas testing instrument, model Witt Oxybaby headspace gas analyzer using an Instron 34SC-5 yniversal tensil testing machine, UK, equipped with a load cell of 5 kN and a crosshead speed of 10 mm.min^−1^ according to ASTM D 882-18. (O_2_/CO_2_) following the method described by [26,27].
P = Q·X/A·tp
where, P is the permeability of gas, (m^3^/m·day·mmHg), Q is the quantity of gas diffused m^3^, X is the thickness of film, A area of the film, m^2^, t is the time, day and ΔP is the pressure difference across the films.


**11. Physico-chemical and microbiological properties:**


**(1). Weight loss:** Weight loss percentage was estimated according to [1]) by using the following equation:$$\mathrm{Weight}\;\mathrm{loss}\;\%=\frac{\mathrm{initial}\;\mathrm{fruit}\;\mathrm{weight}\;-\;\mathrm{fruit}\;\mathrm{weight}\;\mathrm{at}\;\mathrm{sampling}\;\mathrm{date}}{\mathrm{initial}\;\mathrm{fruit}\;\mathrm{weight}}\times 100$$
where g denotes the weight of fruits and vegetables**: the initial weight of fruits.**

**(2). Total soluble solids (TSS):** Total soluble solids were determined by the

Using the Abbe refract meter method at room temperature using an Abbe refract meter (Carl Zeiss Jena) in juice pressed from a sample of slices, according to [1].

**(3). Firmness:** A universal testing machine (cometh, B type, Taiwan) Made in the Taiwan determined texture at the Food Technology Research Institute, Giza, Egypt. provided with software. An aluminium 25 mm diameter cylindrical probe was used in a “Texture Profile Analysis” (TPA) double compression test of penetrating to 50% depth at a one mm/s speed test. Firmness (N), [28].

**(4). Total acidity.** Total acidity was measured for fresh samples as mentioned in the official method of the [29]. It was expressed as citric acid using sodium hydroxide N/10 and phenol phythaline as an indictor.

**(5). Ascorbic acid:** According to, the ascorbic acid content of fresh fruits and vegetables tested was estimated. [29]. using 2,6dichlorophenol-indophenols by

**(6) Anthocyanin:** Total anthocyanin were extracted by adding a solvent containing ethanol HCl (95% ethanol, 1.5 N HCl (85:15)). The solvent was added at level (2:1) solvent to the sample, then the mixture was stored overnight at 4 CO, then filtered on filter paper Atman No.1 and centrifuged at 1000 rpm for 15 min [30]. The supernatant intensity was measured by a spectrophotometer model Spectro UV-Vis 110 V 60 Hz or 220 V 50 Hz Serial No. UV-VIS 0216 Labomed. Inc. (Los Angeles, CA, USA).

**(7). Microbial analysis:** The total microbiological count was determined according to [31]. All the microbial counts were carried out in duplicates.

**(I) Total plate count:** The total colonies of bacteria were estimated using plate count agar medium. The plates were incubated at 37 °C for 48 h.

**(II) Moulds and yeasts count:** The mould and yeast were determined using the methods for the microbiological examination of foods described by the American Public Health Association [32]. by using a malt extract agar medium. The plates were incubated at 25 °C for five days.

**(8) Statistical Analysis:** The collected data was subjected to statistical analysis with the MSTAT statistical software. The mean values were compared using the LSD method at a 5% level. The information was tabulated and statistically analyzed using factorial analysis in a completely randomized design. Experiments were analyzed using the SSPS program according to [33].

## 3. Results and Discussion

The rheological properties such as shear rate, shear stress and viscosity of samples were measured for the prepared nano edible film and coatings at different treatments of A 0%, B 0.02%, C 0.04%, D 0.06%, E 0.02%, F 0.04% and G 0.06% and different shear rates (13.2, 26.4, 39.6, 52.8, 66.00, 79.2 1/s). Figure 1 and Table 2. The results show that the forming solution exhibits non-Newtonian pseudoplastic behaviour at different treatments and fits the power low equation τ = kγn where is the shear stress, Pa is the shear rate per second, k is the consistency index, and n is the flow behaviour index. The results indicated that as the shear rate increased, apparent viscosity decreased. K (consistency index) decreased as the concentration of nano (ZnONPs) edible coating solutions increased (B 0.04%, C 0.08%, and D 0.012%). The same trend was observed in the treatment (E 0.04%, F 0.08% and G 0.12%) of nano (ZnONPs/PPE) edible films and coatings to produce an edible film-forming suspension solution and it was higher than control samples (A 0%), which had a consistency index of (3.69). The flow behaviour index (n) increased with increasing concentration of nano (ZnONPs) edible film and coatings (B 0.04%, C 0.08%, and D 0.012%) to produce an edible film-forming suspension solution and did not show a trend for nano edible film and coatings. It was higher than control with samples A (0%), which had blown behaviour (0.54). This may be due to the effect of the consistency of changes in concentration and the interaction effect between the average value of (k) and (n) for each level of single parameters associated with absolute expression must be eliminated as another function to generate a single parameter model as reported. by [34]. The reaction is interrupted because the particles tend to vibrate at lower and higher temperatures, also splitting larger particles into smaller particles. The results indicated the liquid solution’s behavior, the particles’ type and size, and the presence of nanomaterials and electrolytes. Functionality, constant cost and good availability of polysaccharides such as films and biopolymer films are prerequisites for their use in different industries along with their specialised flow properties The results indicated the relationship between shear rate and shear stress, instantaneous viscosity and shear rate. Graphal representation indicated that the solution exhibited characteristics of a typical non-Newtonian pseudoplastic fluid behavior. The viscosity of each solution showed a high value at the shear rate and decreased linearly with increasing shear stress. [35]. It shows that all samples behave as non-Newtonian pseudoplastic and constitutive equations are necessary to provide the required material parameters by controlling the process, rate of shear stress and properties of mixtures of chitosan components and to determine the relationship between shear time and viscosity for mixing components. Because materials and time depend on mixing, the viscosity decreases continuously over time, which means that when the material is cut, it causes disruption of the collected particles and therefore it provides less resistance to flow and decreases the viscosity over time until the values are stable. He worked on data for the decomposition of shear stress in different treatments and with different concentrations of nanomaterials with chitosan, where it was found that the shear stress variation with shear time was fitted [36].


**The physical and mechanical properties of the prepared nano-edible coating and film solution are combined with ZnONPs and ZnONPs/EPP to produce edible coating and films.**


It can be noticed that the treatment, films prepared nano in both (ZnONPs and ZnONPs/EPP) edible film and coating from the Table 3, the results indicated that the value of thickness values at different treatments was A 0%, B 0.02%, C 0.04%, D 0.06%, E 0.02%, F 0.04%, and G 0.06% gradually increased in thickness with increasing concentration. It was also noted that the nano (ZnONPs) edible film B 0.02% 132, C 0.04%137 and D 0.06%140 um were less in thickness than the concentration of samples following E 0.02%145, F 0.04%160 and G 0.06%175 um nano (ZnONPs/EPP) edible film as compared to the control A 0% thickness value of 186 um, where it was higher than the treatments. It was also observed that the tensile strength in the treatments B, C, and D, 66.22, 34.23, and 48.34 N respectively, was higher than that of the E, F, and G, 54.35, 26.24, and 46.12 N respectively, while the elongation was found to be less in the B, C, and D, 24.69, 23.65, and 22.68% respectively, than that of the E, F, and G, 26.23, 28.35, and 33.18% respectively, as compared with the first initial control samples, the A 0% tensile strength (70.23 N) and elongation (40.45%). When compared to the first initial control samples, the permeability of gases O_2_ and CO_2_ in treatments B, C, and D, 26.84, 39.34, and 43.55, and 40.85, 47.66, and 50.52, respectively, was lower than that of treatments E, F, and G, 40.23, 45.14, and 51.42, and 52.51.The permeability of gases was higher (O_2_, 56.26 M^3^/M^2^10^–7^ day·mmHg and CO_2_, 65.45 M^3^/M^2^10^–8^ day·mmHg).It was also observed that the permeability of water vapor, water vapour transmission rate, and solubility in water were higher in B, C, and D treatments than in other treatments, E, F, and G. As compared with the first initial control samples, it had higher permeability. It was much higher than adding the film ZnONPs, and when GSE was added to ZnONP, the tensile strength was periodically increased, according to [17]. The high elongation of the film is always a desirable characteristic of the film for use in food applications [37]. All the main factors significantly affected the mechanics of the film, as below. The chitosan film had an 18% higher elongation than the chitosan film, and in addition to the incorporation of thymol, this reduced the chitosan layer [25,38]. In general, the as-prepared flake thickness was observed for chitosan-containing films with ZnONPs as plasticizer (B, 132 um) followed by C, (137 um) and (140 um). (145, 160, 175 microns). medium thickness (between 145, 160, and 175 um) as observed in increased ZnONPs/PPE compared to (A, 186 m) structure thickness increased zeta potential [39].


**Determination of particle size distribution and Zeta potentioal produced edible film of the prepared nano-edible coating and films solution combined with ZnONPs and ZnONPs/PPE to produce edible coating and films. The solution formed:**



**Particle size**


The results obtained are shown in Table 2 and Figure 2. At treatments (E20%, F40% and G60%) were higher than the treatments A 0%, B 0.02%, C 0.04%, D 0.06%, E 0.02%, F 0.04% and G 0.06% in both polydisparity index (PdI) and hydrodynamic diameter of particle size (nm). It was found that the peak was (0.648, 0.503, 0.771, and 134, 824.6 and 966 nm) for B 0.04%, C 0.08%, and D 0.12%, respectively. While the peak was E 0.04%, F 0.08% and G 0.12%, the peak was (0.639, 00.650, and 0.503 and 110.6, 188.7, and 873.6 nm) respectively, as compared with initial control samples’ A 0% practical size distribution (nm) in both poly dispersity index (PdI) 0.450 and hydrodynamic diameter of partical size (nm) 1319. A decrease of about 200 nm in the particle size around the charges in the aqueous solution was also observed after the treatment. Where the nano size appeared at around 450 nm, which led to a decrease in the particle size, the zeta potential value decreased. Suspended particles with zeta potentials of above +30 or less than −30 mV repel each other because they are considered stable, but if the zeta potential is between +30 and −30 mV, they tend to attract each other [40,41].


**Zeta potential**


The results show in Table 4 and Figure 3 that the Zeta potential distribution and Zeta Deviation (mV) were recorded for the treatment in the peak was (52.4, 36.5 and 29.5) and (4.17, 3.61 and 3.20) for B 0.02%, C 0.04% and D 0.06% respectively. While it was found that the figure recorded that the concentration of the treatments’ Zeta potential distribution and Zeta Deviation (mV), E 0.02%, F 0.04% and G 0.06% were recorded for the treatment in the peak was (39.2, 42.5 and 40.9) and (3.89, 3.81 and 3.61) respectively, as compared with initial control samples, A 0% in both Zeta potential distribution (49.3 and Zeta Deviation (mV) 5.12 (mV). The zeta potential measures the electric charge at the colloidal particle boundary and is an important indicator of the surface charges. The zeta potential determines the electrostatic repulsion between them and is responsible for their stability against sedimentation. It is usually obtained by measurement and conversion from the electrical kinetics of particles, tests, and materials (ASTM 2003). A zeta potential with an absolute value greater than 30 mV indicates “medium to good” colloidal systems (ASTM) stability. The higher the zeta potential, the better the dispersion stability. Desirable zeta potentials can be obtained by [42,43]. The reason for the increase in zeta potential may be due to the fact that the components of the chitosan complex polymer with glacial acetic acid and PPE extracts, 5%, were the reason for the increase. Also, the reason for the increase in zeta potion may be due to the immiscibility of the solution, and this appeared in the electron microscope with a crack in the control film. As seen in the increased ZnO NPs/PPE in comparison to (A, 186 m), The thickness of the structure increased the zeta potential [44].


**Scanning electron microscopy (SEM) microstructure of prepared nano edible coatings and films combined with ZnONPs and ZnONPs/EPP to produce edible films:**


There are seven microscopic images prepared nano edible coatings and films combined with ZnONPs and ZnONPs/EPP to produce edible film. After taking the cross-section, the morphology of the surface was investigated using SEM images (Figure 4), and the surface roughness was estimated using nano-edible coatings and films. The SEM images showed that the edible films were intact and smooth without any noticeable delamination on the surface, which was found to rise with increasing concentration upon different treatments In the figure of prepared nano-edible coatings and films combined with ZnONPs and ZnONPs/EPP to produce edible films, A 0%, B 0.02%, C 0.04%, D 0.06%, E 0.02%, F 0.04%, and G 0.06% are shown. It was found that the control, a 0% treatment, has cracks in the edible film due to poor treatment that does not contain ZnONPs and ZnONPs/EPP. The treatments in the picture also showed that the addition of ZnONPs and ZnONPs/EPP characteristics produces edible films with a smooth surface flush D with some rough ridges, and treatment G found roughness of transparent appearance and droplets of elliptical shape. Treatment B was homogeneous with bubble drops compared to a spherical morphological droplet. The edible film contains a homogeneous solution with some compact fine grains and an intact smooth crystal morphology in a continuous matrix in general. It can be concluded that these studies are useful for learning about the microstructure and membrane morphology, which can help in selecting the edible nanofilm formulas for coating and packing. Our results are in agreement with those obtained by [38]). SEM images found that the elegant CMC film was smooth and intact without any deformation, and the surface was clear. The surface roughness was slightly increased in the films supplemented with grape seed extract, which was confirmed by the higher surface roughness of the CMC/grape seed extract film compared to the elegant CMC film [14,45]. The presence of a gelatinous mass with cracking in the control film was observed, which may be due to the incompatibility of the complex polymer with pomegranate peel extract. Generally, the scanning microscopy study may be useful for recognizing the microstructure and morphology of the produced films, which can be helpful in choosing the proper film formula for coating and packaging. Purposes. Also, the color appearance of produced films may be important because it could affect consumer acceptance of coated items [33].


**Determination of colour apparent & transmittance light of the edible films of prepared nano-edible coatings and films combined with ZnONPs and ZnONPs/PPE to produce edible films using scanning electron microscopy (SEM) technology:**


The results obtained are presented in Table 5 and Figure 5 and Figure 6. According to the curves in figure, the apparent colour increased with increasing concentration at different treatments: A 0%, B 0.02%, C 0.04%, D 0.06%, E 0.02%, F 0.04%, and G 0.06% of prepared nano edible coatings and films combined with ZnONPs and ZnONPs/PPE to produce edible films.It was also noted that the treatments were higher value of apparent colour was recorded for the treatment, B0.04%, 64.51, 0.42, 6.49 and 31.01 C 0.08%, 62.45, 0.60, 6.36 and 27.65, D 0.12%, 61.48, 0.75, 7.05 and 24.205 for L, a, b and E respectively, while that the treatment, E 0.04%, 62.44, 0.51, 7.50 and 26.70, F 0.08%, 63.40, 0.76, 10.72 and 30.58, G 0.12%, 61.20, 0.64, 12.74 and 26.62 for L, a, b and E respectively as compared A 0% for L, a, b and E was 56.50, 0.22, 7.04 and 56.56. On the other hand, the barrier and transparency of the films were evaluated by measuring the percent light transmittance at T550, 430, and 395 nm, respectively. It was found that the treatments decreased with increasing concentration at different treatments in both B 0.04%, C 0.08%, and D 0.12% and E 0.04%, F 0.08%, and G 0.12% compared to the control A 0% sample, and the results were as follows: B20%, 73.1, 62.2, 53.7, and C40%, 65.9, 52.3, 43.2, and D60%, 54.7, 44.2, and 37.3 for T550, 430, and 395 nm respectively, while the treatment, E20%, 53.4, 39.4, 28.4, and F40%, 61.6, 45.4, and G60%, 64.6, 54.4, and 63.4 for T550, 430, and 395 nm respectively. Interestingly, the addition of nanomaterial did not significantly reduce the transparency of the film (by less than 10%) but significantly reduced the light transmittance depending on the concentration of nanomaterial. When two wt% of nanomaterial were added, the T280 of the film was reduced by more than 50%. These results indicate the addition of nanomaterial significantly increased the UV barrier property without sacrificing the transparency of the film. The UV blocking property of pectin/agar films containing nanomaterial is mainly attributed to the UV light absorption function of the nanomaterial. [21,23,37,46].


**Applications of chosen proper nanomaterial’s of edible films to manfalouty pomegranate fruit:**



**Physico-chemical and microbiological of coated manfalouty pomegranate fruit during storage period.**


### 3.1. Weight Loss Percentage

The results obtained are presented in Table 6. From this table, it could be observed that the weight loss increased with increasing the storage period at a cold temperature for treatments A 0%, B 0.02%, C 0.04%, D 0.06%, E 0.02%, F 0.04%, and G 0.06% packaged in nano-coated. Compared to the control, A0% indicated a higher weight loss than the nano-coated treatments. It was also noted that the higher the percentage of treatment concentration, the greater the weight loss in nano-coated treatments. It was also pointed out that the treatments ZnONPs/PPE were better than ZnONPs in reducing the weight loss percentage compared to the control sample, which was much higher. The results of the study showed that the best treatment was G, followed by D, followed by F, C, E, and B, as compared with control A0%, which was the least. Typically, weight loss occurs during fruit storage due to its respiratory process, the transference of humidity, and some oxidation processes [47].This weight loss is correlated with an increase in water loss due to a rise in transpiration and respiration [48].the high surface/volume ratio explains the sweet cherries and their low skin diffusion resistance [49].Similar results were reported in previous studies with other edible coatings and concluded that edible films created a physical barrier against moisture loss and reduced the transpiration rate [50,51].

### 3.2. Total Soluble Solids (TSS%)

Changes occurring in TSS% of edible coatings to pomegranate in Table 7, it could be noticed that TSS% of nano-coated pomegranate gradually decreased with increasing the storage period at cold temperatures in both treatments ZnONPs/PPE and ZnONPs. It was also found that the percentage of TSS decreased in treatments ZnONPs/PPE than in treatments ZnONPs compared to the control A% sample, where the rate of total solids loss was much higher than the coated treatments. This reduction might be due to the respiratory process. All treatments significantly reduced the loss of TSS%, except coating treatment G, which recorded the highest values (12.74%). Coating film on the surface of fruits reduces respiration rate and vital processes, which reduces the loss of TSS% during storage. The total soluble solids of uncoated and coated fruits after 20 days of storage. This value increased slightly during storage, probably due to the water loss, activity of hydrolytic enzymes, or the decrease in respiration rate and conversion of sugars into CO_2_ and H_2_O during the storage period [50,51].

### 3.3. Total Acidity

The changes in total acidity of pomegranate were determined during the storage period, i.e., The obtained results are recorded in Table 8. The results indicated that the total acidity gradually decreased with increasing nano-coated pomegranate during the storage period. Total acidity was significantly reduced as a function of storage time for all studied treatments. It shows the delay in using organic acid in enzymatic respiration. All treatments showed significant differences at the end of storage; treated arils by ZnONPs carried by chitosan or chitosan ZnONPs/PPE recorded the highest values of acidity compared with other treatments and control. It is also considered that coatings reduce the respiration rate and may delay the utilisation of organic acids. Retention of acidity has been reported previously for various strawberries treated with edible coatings and films. [20]. The acidity of uncoated and coated fruits after 20 days of storage Acidity estimates the fruit’s organic acid content, which generally decreases during postharvest storage due to the use of organic acids as substrates for respiratory metabolism [52]. Acidity decreased during storage time in uncoated and coated fruit samples, but this decrease was less noticeable in coated fruit [51].

### 3.4. Ascorbic Acids

The obtained results are recorded in Table 9; the results indicate that the ascorbic acid decreased with increasing nano-coated pomegranate during storage. It was also found that the percentage of ascorbic acid decreased in treatments ZnONPs/PPE than in the treatments ZnONPs compared to the control A% sample, where the rate of ascorbic acid loss was much higher than in the coated treatments. The decrease in ascorbic acid during storage may be due to the conversion of ascorbic acid to dehydroascorbic acid due to the action of ascorbic acid oxidase. Ascorbic acid is a potent free radical scavenger that prevents fruit degradation during ripening. The ascorbic acid content had no significant difference (*p* > 0.05) between different samples during the storage time. In contrast, it was previously found that applying chitosan coatings enriched with pomegranate peel extract on guava decreased the loss of ascorbic acid during storage. Similar results of delays in ascorbic acid in chitosan-coated sweet cherries were reported [50].Who hypothesized that coating materials lowered the oxygen permeability and the activity of enzymes, resulting in the prevention of the oxidative deterioration of ascorbic acid [51].

### 3.5. Anthocyanin Percentage

Data in Table 10 shows the effect of ZnONPs and chitosan combined with ZnONPs/PPE on the percentage of anthocyanin in Manfalouty pomegranate arils stored for 5, 10, 15, and 20 days at 2 °C during storage days. The anthocyanin content of arils decreased during storage periods. However, after ten days of storage, results indicated highly significant differences between all treatments. ZnONPs carried by chitosan recorded the highest value (14.73%), followed by ZnONPs/PPE (14.62%). But after 20 days of storage periods, the data showed no significant differences. [6] noted that the essential oils had the lowest anthocyanin value. Anthocyanins, one of the major flavonoids, are natural water-soluble pigments connected with the ripening stage of the fruit and are mainly used as an indicator of cherry quality. The concentration and distribution of cyanidin-3-O-glucoside and cya-indin-3-O-rutinoside in the skin influence the colour of cherries. As can be seen in Figure 3, anthocyanins increased during the storage period as ripening progressed. After harvest, the total anthocyanin content was 23–65 mg/100 g. A significant difference “*p* ≥ 0.05” in the amounts of anthocyanin was related to chitosan enriched with olive leaf extract and control samples. An increase in anthocyanins after harvest has been previously reported for other fruits such as strawberries and pomegranates and may be related to the anthocyanin synthesis in the fruit during storage. [50].

### 3.6. Total Microbial Count

The data indicates that total counts gradually increased with increasing the cold storage period of pomegranate in both ZnONPs and ZnONPs/PPE of edible film nano-materials. The results in Table 11 showed that the microbial load increases with the increase of storage in all storage treatments. Control A% treatment indicates higher counts than the coated nanomaterials. The growth of microorganisms increased with a prolonged storage period, particularly in the untreated arils. All types had lower levels of microbial lead in comparison to control. The data revealed highly significant differences among all treatments, whatever the storage period. Ten days of storage time were squandered.ZnONPs/PPE combined with chitosan recorded the lowest value of 3.12 × 10^1^ CFU/g, followed by ZnONPs combined with chitosan at 3.36 × 10^1^ CFU/g. After 20 days of storage Adding nanoparticle material as an antimicrobial agent to coating emulsion improved the fruit’s microbial quality. It decreased microbial counts [53]. It found that the coating treatment of fruit and vegetables allowed limited gas exchange and respiration, preventing the fermentation process and minimising the microbial count. Also, they added that high microbial counts appeared after three weeks in control samples stored at room temperature. In addition, it was remarked that samples treated with non-nano coating showed higher counts after storage than those treated with nano coating. This may be due to the wax’s relatively high thickness, which prevents respiration and leads to anaerobic conditions and fruit degradation. In addition, it could be reported that the edible coating and film nanoparticle material have a remarkable effect on the rate of microbial counts during storage at a cooled temperature [52].

### 3.7. Mold and Yeast

The results are shown in Table 12 Mold and yeast counts gradually increased with increasing storage time at a cooled temperature.The data revealed highly significant differences among all treatments, whatever the storage period. Ten days of storage time were squandered.ZnONPs combined with chitosan recorded the lowest value of 2.75 × 10^1^ CFU/g, followed by ZnONPs/PPE combined with chitosan, 2.46 × 10^1^ CFU/g. After 20 days of storage. Edible coatings with 0.2% ZnO2 were the most effective, decreasing yeast and mould growth at 6 and 12 days, though the bacterial load increased after 12 days of storage. The combination of CMC with nano-ZnO2 helped maintain bioactive compounds in the ready-to-eat pomegranate [31]. Who noted that the essential oils reduced yeast and moulds colony [13].on orange fruit stated that essential oils did not control the producing agent of the blue fungus. In addition, bio-films and coatings act as carriers of food additives (i.e., antioxidants and antimicrobials) and have been mainly considered in food preservation due to their ability to extend the shelf life.

### 3.8. Color Changes

The colour changes were measured by recording lightness (L* value), chroma (intensity of color) and hue angle (*h◦*). The lightness of the strawberries was affected by storage time (Table 13). The results indicated that the manfalouty pomegranate gradually decreased with an increasing storage period at cooling temperatures. The lightness (L*) gradually decreased during storage in both ZnONPs and ZnONPs/PPE manfalouty pomegranates. The highest decrease in lightness was observed in uncoated (control, A%) manfalouty pomegranates. Changes in the hue-angle (*h◦*) value of coated fruit with storage time were slight and only became significant at the end of the storage period. Chroma was reduced by around 30% for control and 10% for coated fruit. The coating solution gave rise to significant differences in fruit colour by the end of the storage period and the surface colour of fruits [20].

## 4. Conclusions

Physical and chemical properties were studied, e.g., rheological properties and paratical size distribution, zeta potential, and scanning electron microscopy films. Adding nanomaterials to edible coating prolongs product shelf life, reduces the risk of pathogen growth, and improves the quality of fruit and vegetable surfaces by using low-cost substrates of nano-polymer production to produce edible coating and film solutions. As this substance inhibits microbial growth and oxidation and improves the quality of the film, polysaccharides are extracted from polysaccharides to preserve the film from oxidation and microbial contamination, maintain the stability of the film, and extend the shelf life of food; acceptable sensory characteristics; appropriate barrier properties (CO_2_, O_2_, and water); microbial biochemical and physic-chemical stability; safety and health; effective carrier for antioxidants was found that the best measurements were that the film-forming nan emulsion solutions decreased by E% 110 nm and B% 134 nm. Nano followed treatment, F% 188.7 nm, compared to nano edible films, which were A0% 1312nm. Acceptable sensory properties are those suitable for barrier properties (carbon dioxide, oxygen, and water). Thus, the stability and efficiency of these barriers to carrying antioxidant additives could indicate that these results may be useful in the food industry domain and simple technology for the low-cost production process.

## Figures and Tables

**Figure 1 materials-16-01569-f001:**
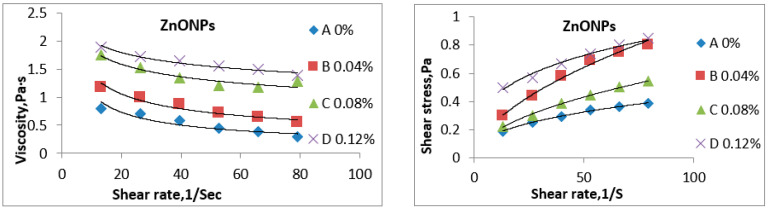
Shear rate, shear stress, and viscosity of nanoedible film to produce an edible film-forming suspension solution of different treatments.

**Figure 2 materials-16-01569-f002:**
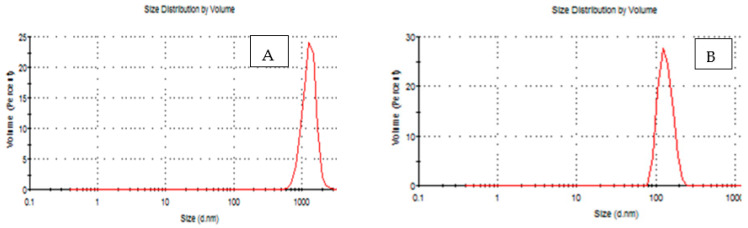
Partical size of prepared nano-edible coatings and films combined with ZnONPs and ZnONPs/EPP to produce an edible film-forming suspension solution of different treatments. (**A**) = PPE extracts, 5% (0.1%) wt% chitosan (**B**) = ZnONPs, 1% (0.02%) wt% chitosan (**C**) = ZnONPs, 2% (0.04%) chitosan (**D**) wt% = ZnONPs, 3% (0.06%) wt% chitosan weight (**E**) = ZnONPs 1%/PPE1% (0.02%) ZnONPs 2%/PPE2% (0.04%) chitosan weight (**F**) chitosan weight (**G**) = ZnONPs 3%/PPE3% (0.06%) Chitosan weight n.s. stands for “not significantly”.

**Figure 3 materials-16-01569-f003:**
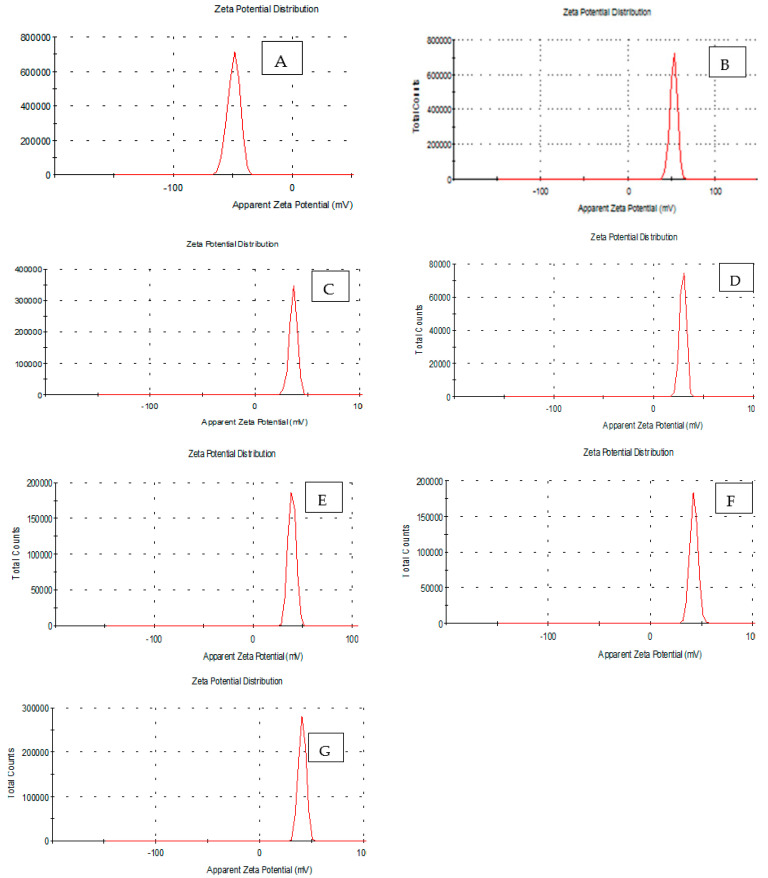
Zeta potentioal of prepared nano edible coatings and films combined with ZnONPs and ZnONPs/EPP to produce an edible film-forming suspension solution of different treatments. (**A**) = PPE extracts, 5% (0.1%) wt% chitosan (**B**) = ZnONPs, 1% (0.02%) wt% chitosan (**C**) = ZnONPs, 2% (0.04%) chitosan (**D**) wt% = ZnONPs, 3% (0.06%) wt% chitosan weight (**E**) = ZnONPs 1%/PPE1% (0.02%) ZnONPs 2%/PPE2% (0.04%) chitosan weight (**F**) chitosan weight (**G**) = ZnONPs 3%/PPE3% (0.06%) Chitosan weight n.s. stands for “not significantly”.

**Figure 4 materials-16-01569-f004:**
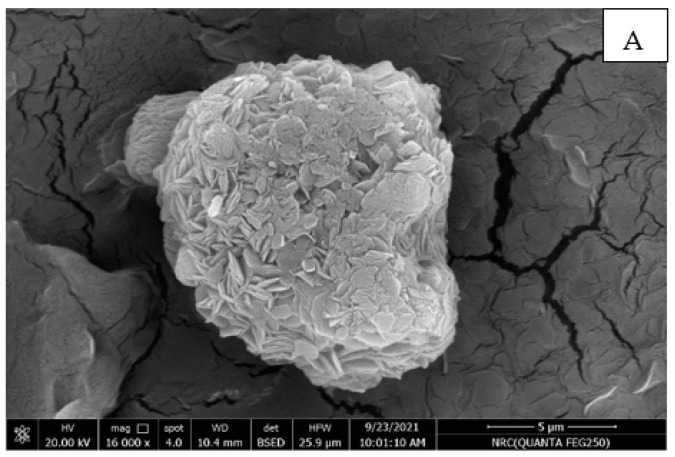
Microstructure of prepared nano edible coatings and films combined with ZnONPs and ZnONPs/EPP to produce edible films using scanning electron microscopy (SEM) technology using scanning electron microscopy (SEM) at different treatments. (**A**) = PPE extracts, 5% (0.1%) wt% chitosan (**B**) = ZnONPs, 1% (0.02%) wt% chitosan (**C**) = ZnONPs, 2% (0.04%) chitosan (**D**) wt% = ZnONPs, 3% (0.06%) wt% chitosan weight (**E**) = ZnONPs 1%/PPE1% (0.02%) ZnONPs 2%/PPE2% (0.04%) chitosan weight (**F**) chitosan weight (**G**) = ZnONPs 3%/PPE3% (0.06%) Chitosan weight n.s. stands for “not significantly”.

**Figure 5 materials-16-01569-f005:**
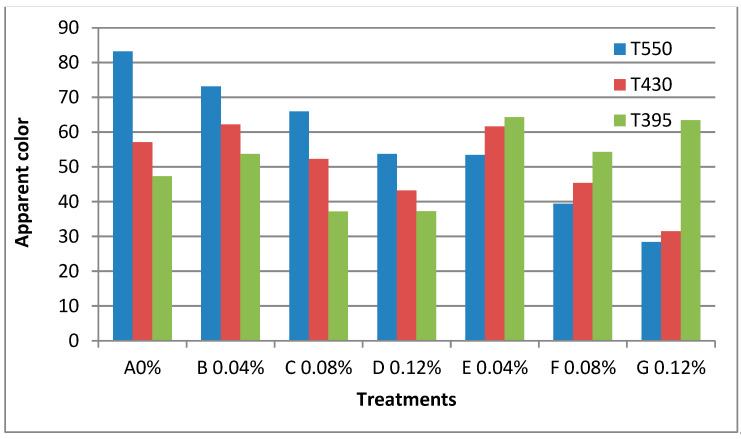
Measured of apparent color (Light transmittance) of the edible films prepared nano edible coating & films solution combined with ZnONPs and ZnONPs/PPE on edible films at different treatments. A 0%, B 0.04%, C 0.08%, D 0.12%, E 0.04%, F 0.08% and G 0.12%.

**Figure 6 materials-16-01569-f006:**
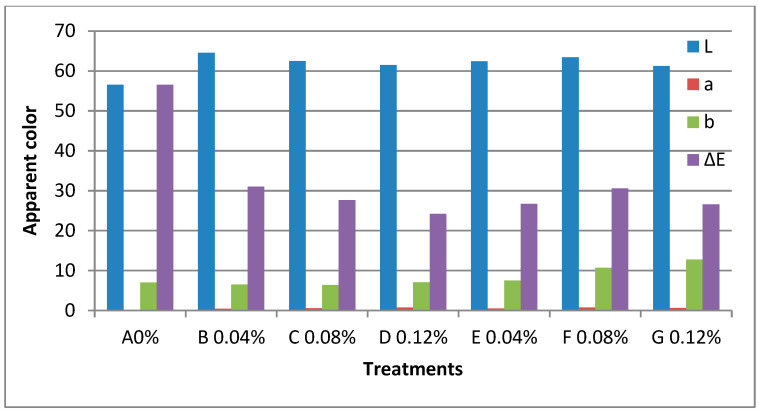
Measured of apparent color (Minolta Chroma) of the edible films prepared nano edible coating & films solution combined with ZnONPs and ZnONPs/PPE on edible films at different treatments A 0%, B 0.04%, C 0.08%, D 0.12%, E 0.04%, F 0.08% and G 0.12%.

**Table 1 materials-16-01569-t001:** The table shows the composition of the solubility chitosan film in glacial acetic acid, plasticizers such as glycerol, coefficients code, and permissible percentages of zinc oxide and pomegranate peel extracts.

		Films Components %				
Film Code	Chitosan	Glacial Acetic Acid	Glycerol	Pomegranate Peel Extracts (PPE) 5% (0.1%)	ZnONPs	ZnONPs/PPE
A 0%	2%	2 mL	1.2%	0.1%	-	-
B 0.02%	2%	2 mL	1.2%	-	1% (0.02%)	-
C 0.04%	2%	2 mL	1.2%	-	2% (0.04%)	-
D 0.06%	2%	2 mL	1.2%	-	3% (0.06%)	-
E 0.02%	2%	2 mL	1.2%	-	-	1% (0.02%)/0.1%
F 0.04%	2%	2 mL	1.2%	-	-	2% (0.04%)/0.1%
G 0.06%	2%	2 mL	1.2%	-	-	3% (0.06%)/0.1%

(A) = PPE extracts, 5% (0.1%) chitosan (B) wt% = ZnONPs, 1% (0.02%) wt% chitosan (C) = ZnONPs, 2% (0.04%) ZnONPs, 3% (0.06%) chitosan (D) wt% chitosan weight (E) = ZnONPs 1%/PPE1% (0.02%) ZnONPs 2%/PPE2% (0.04%) chitosan weight (F) chitosan weight (G) = ZnONPs 3%/PPE3% (0.06%) chitosan weight.

**Table 2 materials-16-01569-t002:** Shows the relationship between the flow behaviour index (n) and the consistency index (k) at various treatments. Preparation of prepared chitosan nanofilm solution combined with ZnONPs and ZnONPs/PPE to produce an edible film-forming suspension solution: A 0%, B 0.04%, C 0.08%, D 0.12%, E 0.04%, F 0.08%, and G 0.12%.

Treatments	Viscosity, cp			Shear Stress, Pa		
	k	n	R^2^	k	n	R^2^
A 0%	3.697	0.541	0.879	0.0674	0.4021	0.997
Preparation of nano films solution combined with ZnONPs to produce edible film-forming suspension solution						
B 0.02%	3.645	0.414	0.943	0.0705	0.5648	0.9935
C 0.04%	3.046	0.218	0.007	0.0586	0.5102	0.9972
D 0.06%	2.9136	0.162	0.0963	0.2199	0.3058	0.9816
Preparation of nano films solution combined with ZnONPs/PPE to produce edible film-forming suspension solution						
E 0.02%	1.239	0.616	0.9881	0.3516	0.4662	0.9852
F 0.04%	23.001	0.542	0.9052	0.0678	0.8404	0.9997
G 0.06%	1.5675	0.533	0.9435	0.0253	0.9579	0.9985
L.S.D	0.389	0.148	0.353	1.90	0.152	n.s
Standard error	0.1285	4.8870	0.1164	6.281	5.0185	0.04312
Mean	-	-	-	-	-	-
A	3.695	0.5503	0.885	0.0994	0.43803	0.8943
B	3.645	0.4340	0.939	0.9716	0.59893	0.9415
C	3.123	0.2206	0.140	0.1933	0.5780	0.9604
D	2.91	0.2330	0. 390	0.2296	0.3319	0.9452
E	1.260	0.6483	0.9620	0.3402	0.5104	0.9204
F	23.310	0.546	0.8960	0.0681	0.8651	0.9582
G	1.59	0.5483	0.9328	0.0909	0.9439	0.5878

(A) = PPE extracts, 5% (0.1%) wt% chitosan (B) = ZnONPs, 1% (0.02%) wt% chitosan (C) = ZnONPs, 2% (0.04%) chitosan (D) wt% = ZnONPs, 3% (0.06%) wt% chitosan weight (E) = ZnONPs 1%/PPE1% (0.02%) ZnONPs 2%/PPE2% (0.04%) chitosan weight (F) chitosan weight (G) = ZnONPs 3%/PPE3% (0.06%) chitosan weight (n.s. = not statistically significant).

**Table 3 materials-16-01569-t003:** Mechanical properties, permeability, and thickness of the prepared nano-edible coating and film solution in combination with ZnONPs and ZnONPs/EPP to produce edible coating and films.

Treatments	Thickness Um	Tensile Strength N	Elongation %	O_2_ M^3^M/M^2^ × 10^−7^Day·mmHg	CO_2_ M^3^·M/M^2^ × 10^−8^Day·mmHg	Water Vapors Transmission Rate, g/h·m^2^	Water Vapors g·mm/m^2^·Day·mmHg	Solubility in Water %
A 0%	186	70.23	40.45	56.26	65.45	45	0.1707	80.34
produced edible film of prepared nano edible coating & films combined with ZnONPs to produce edible coating and films								
B 0.02%	132	66.22	24.69	36.84	40.85	19.94	0.1342	65.27
C 0.04%	137	34.23	23.65	39.34	47.66	27.63	0.1430	67.34
D 0.06%	140	48.34	22.68	43.55	50.52	33.44	0.1538	70.78
produced edible film of prepared nano edible coating & films combined with ZnONPs and ZnONPs/PPE to produce edible coating and films								
E 0.02%	145	54.35	26.23	40.23	52.51	16.62	0.1315	29.18
F 0.04%	160	26.24	28.35	45.14	56.77	19.35	0.1240	25.47
G 0.06%	175	46.12	33.18	51.42	58.64	24.28	0.1138	22.28
L.S.D	1.75	1.73	1.76	1.70	1.77	1.72	n.s	8.927
Standard error	0.5773	0.5710	0.5803	0.5637	0.5864	0.568	4.178	2.9433
Mean	-	-	-	-	-	-	-	-
A	186	70.56	40.51	83.5	65.62	45	0.1640	80.70
B	132	66.61	24.59	72.5	40.71	19.78	0.1519	65.63
C	137	34.62	23.57	65.76	47.65	27.51	0.1585	67.59
D	140	48.62	22.66	54.70	50.57	33.50	0.1620	70.84
E	145	54.55	26.47	52.56	52.49	16.60	0.1540	29.60
F	160	26.99	28.58	61.56	56.69	19.51	0.1520	25.66
G	175	46.59	33.38	64.53	58.51	24.45	0.1449	15.33

(A) = PPE extracts, 5% (0.1%) wt% chitosan (B) = ZnONPs, 1% (0.02%) wt% chitosan (C) = ZnONPs, 2% (0.04%) chitosan (D) wt% = ZnONPs, 3% (0.06%) wt% chitosan weight (E) = ZnONPs 1%/PPE1% (0.02%) ZnONPs 2%/PPE2% (0.04%) chitosan weight (F) chitosan weight (G) = ZnONPs 3%/PPE3% (0.06%) Chitosan weight n.s. stands for “not significantly”.

**Table 4 materials-16-01569-t004:** Measured particle size and zeta potentioal of nanotechnology of prepared nano-edible coating and film solutions combined with ZnONPs and ZnONPs/PPE to produce edible coating and film solutions formed from it.

Treatments	Partical Size Distribution (nm)		Zeta Potential (mv)	
	Poly Dispersity Indexs Pdi	Hydrodynamic Diameter nm	Z-Potentioal	Z-Devition
A 0%	0.450	1312	−49.3	5.12
produced edible film of prepared nano edible coating & films solution combined with ZnONPs to produce edible coating and films				
B 0.02%	0.648	134	52.4	4.17
C 0.04%	0.503	824.6	36.61	3.61
D 0.06%	0.771	966	29.5	3.20
produced edible film of prepared nano edible coating & films solution combined with ZnONPs and ZnONPs/PPE to produce edible coating and films				
E 0.02%	0.639	110.6	39.2	3.89
F 0.04%	0.650	188.7	42.5	3.81
G 0.06%	0.503	873.6	40.9	3.61
L.S.D	8.01	31.08	18.61	0.455
Standard error	0.02641	10.2470	6.138	0.150
Mean	-	-	-	-
A	0.45	1309	32.866	5.19
B	0.65	134	52.46	4.31
C	0.50	824	36.60	3.51
D	0.81	966	29.56	3.26
E	0.63	136.86	38.35	3.37
F	0.65	186	42.60	3.80
G	0.53	8.75.2	40.77	3.49

(A) = PPE extracts, 5% (0.1%) wt% chitosan (B) = ZnONPs, 1% (0.02%) wt% chitosan (C) = ZnONPs, 2% (0.04%) chitosan (D) wt% = ZnONPs, 3% (0.06%) wt% chitosan weight (E) = ZnONPs 1%/PPE1% (0.02%) ZnONPs 2%/PPE2% (0.04%) chitosan weight (F) chitosan weight (G) = ZnONPs 3%/PPE3% (0.06%) Chitosan weight n.s. stands for “not significantly”.

**Table 5 materials-16-01569-t005:** Measured of color apparent and transmittance light transmittance of prepared nano edible coating & films combined with ZnONPs and ZnONPs/EPP to produce edible films using scanning electron microscopy (SEM) technology.

Films	L	a	b	ΔE	T550 nm	T430 nm	T395 nm
A 0%	56.5	0.22	7.04	56.94	83.2	57.1	47.3
produced edible film of prepared nano edible coating & films solution combined with ZnONPs to produce edible coating and films							
B 0.02%	64.51	0.42	6.49	31.01	73.1	62.2	53.7
C 0.04%	62.45	0.60	6.36	27.65	65.9	52.3	43.2
D 0.06%	61.48	0.75	7.05	24.25	54.7	44.2	37.3
produced edible film of prepared nano edible coating & films solution combined with ZnONPs and ZnONPs/PPE to produce edible coating and films							
E 0.02%	62.44	0.51	7.50	26.70	53.4	39.4	28.4
F 0.04%	63.40	0.76	10.72	30.58	61.6	45.4	31.5
G 0.06%	61.20	0.64	12.74	26.62	64.3	54.3	63.4
L.S.D	1.643	0.0015	0.487	1.748	1.709	1.787	1.509
Standard error	0.5418	0.024	0.1606	0.5766	1.709	0.5893	0.4976
Mean	-	-	-	-	-	-	-
A	55.41	0.23	7.06	55.86	83.5	58	47.6
B	64.67	0.40	6.573	30.66	72.5	62.53	53.73
C	62.55	0.62	6.233	27.69	65.76	51.53	42.56
D	61.41	0.74	6.983	24.55	54.70	44.6	37.56
E	62.58	0.50	7.343	25.66	52.65	93.66	28.46
F	63.46	0.73	10.496	30.73	61.65	45.66	31.33
G	61.36	0.61	12.13	25.67	64.53	54.60	63.56

(A) = PPE extracts, 5% (0.1%) wt% chitosan (B) = ZnONPs, 1% (0.02%) wt% chitosan (C) = ZnONPs, 2% (0.04%) chitosan (D) wt% = ZnONPs, 3% (0.06%) wt% chitosan weight (E) = ZnONPs 1%/PPE1% (0.02%) ZnONPs 2%/PPE2% (0.04%) chitosan weight (F) chitosan weight (G) = ZnONPs 3%/PPE3% (0.06%) Chitosan weight n.s. stands for “not significantly”.

**Table 6 materials-16-01569-t006:** Effect of edible film and coatings on weight loss (%) of manfalouty pomegranate during storage at 2 °C.

Treatments	A 5% (0.1%)	ZnONPs			ZnONPs/PPE		
		B 1% (0.02%)	C 2% (0.04%)	D 3% (0.06%)	E 1% (0.02%)	F 2% (0.04%)	G 3% (0.06%)
Zero time	-	-	-	-	-	-	-
5	2.25	2.20	1.90	1.71	2.10	1.93	1.82
10	3.89	2.50	1.98	1.87	2.45	2.04	1.89
15	-	2.85	2.25	1.92	2.76	2.15	1.90
20	-	2.90	2.50	1.99	2.88	2.45	1.95
L.S.D		S = 1.15		T = 1.54		S&T = 0.18	

(-) a spoiled reject samples. LSD Treatments = T LSD Storage period = S LSD (Storage period ∗ Treatments) = T ∗ S. Means within a column showing the same letters are not significantly different (*p* ≥ 0.05). (A) = PPE extracts, 5% (0.1%) wt% chitosan (B) = ZnONPs, 1% (0.02%) wt% chitosan (C) = ZnONPs, 2% (0.04%) wt% chitosan (D) = ZnONPs, 3% (0.06%) wt% chitosan (E) = 1% ZnONPs/1% PPE (0.02%) wt% of chitosan (F) = ZnONPs 2%/PPE2% (0.04%) wt% of chitosan (G) = ZnONPs 3%/PPE3% (0.06%). wt% of chitosan.

**Table 7 materials-16-01569-t007:** Effect of edible film and coatings on total soluble solids (TSS) of manfalouty pomegranate during storage at 2 °C.

Treatments	A (PPE), 5% (0.1%)	ZnONPs			ZnONPs/PPE		
		B 1% (0.02%)	C 2% (0.04%)	D 3% (0.06%)	E 1% (0.02%)	F 2% (0.04%)	G 3% (0.06%)
Zero time	15.75	15.75	15.75	15.75	15.75	15.75	15.75
5	14.56	15.69	15.62	15.58	15.73	15.60	15.53
10	12.10	14.85	14.44	14.38	14.81	14.41	14.34
15	-	13.68	13.35	13.28	13.63	13.32	13.25
20	-	12.15	12.20	12.49	12.22	12.57	12.76
L.S.D		S = 1.11		T = 1.45		S&T = 0.21	

(-) a spoiled reject samples. LSD Treatments = T LSD Storage period = S LSD (Storage period ∗ Treatments) = T ∗ S. Means within a column showing the same letters are not significantly different (*p ≥* 0.05). (A) = PPE extracts, 5% (0.1%) wt% chitosan (B) = ZnONPs, 1% (0.02%) wt% chitosan (C) = ZnONPs, 2% (0.04%) wt% chitosan (D) = ZnONPs, 3% (0.06%) wt% chitosan (E) = 1% ZnONPs/1% PPE (0.02%) wt% of chitosan (F) = ZnONPs 2%/PPE2% (0.04%) wt% of chitosan (G) = ZnONPs 3%/PPE3% (0.06%). wt% of chitosan.

**Table 8 materials-16-01569-t008:** Effect of edible film and coatings on acidity of manfalouty pomegranate during storage at 2 °C.

Treatments	A(PPE), 5% (0.1%)	ZnONPs			ZnONPs/PPE		
		B 1% (0.02%)	C 2% (0.04%)	D 3% (0.06%)	E 1% (0.02%)	F 2% (0.04%)	G 3% (0.06%)
Zero time	0.81	0.81	0.81	0.81	0.81	0.81	0.81
5	0.80	0.83	0.82	0.84	0.82	0.81	0.84
10	0.78	0.82	0.81	0.80	0.81	0.80	0.79
15	-	0.80	0.79	0.79	0.79	0.78	0.77
20	-	0.79	0.78	0.75	0.78	0.76	0.74
L.S.D		S = 1.15		T = 1.39		S&T = 0.14	

(-) a spoiled reject samples. LSD Treatments = T LSD Storage period = S LSD (Storage period ∗ Treatments) = T ∗ S. Means within a column showing the same letters are not significantly different (*p* ≥ 0.05). (A) = PPE extracts, 5% (0.1%) wt% chitosan (B) = ZnONPs, 1% (0.02%) wt% chitosan (C) = ZnONPs, 2% (0.04%) wt% chitosan (D) = ZnONPs, 3% (0.06%) wt% chitosan (E) = 1% ZnONPs/1% PPE (0.02%) wt% of chitosan (F) = ZnONPs 2%/PPE2% (0.04%) wt% of chitosan (G) = ZnONPs 3%/PPE3% (0.06%). wt% of chitosan.

**Table 9 materials-16-01569-t009:** Effect of edible film and coatings on ascorbic acid of manfalouty pomegranate during storage at 2 °C.

Treatments	A(PPE), 5% (0.1%)	ZnONPs			ZnONPs/PPE		
		B 1% (0.02%)	C 2% (0.04%)	D 3% (0.06%)	E 1% (0.02%)	F 2% (0.04%)	G 3% (0.06%)
Zero time	17.50	17.50	17.50	17.50	17.50	17.50	17.50
5	12.45	13.62	13.10	12.37	13.54	13.01	12.13
10	11.18	13.40	12.89	11.86	13.22	12.65	11.49
15	-	12.13	12.48	11.28	12.88	12.25	11.42
20	-	12.26	11.32	11.05	12.63	11.86	11.26
L.S.D		S = 1.15		T = 1.56		S&T = 0.15	

(-) a spoiled reject samples. LSD Treatments = T LSD Storage period = S LSD (Storage period ∗ Treatments) = T ∗ S. Means within a column showing the same letters are not significantly different (*p ≥* 0.05). (A) = PPE extracts, 5% (0.1%) wt% chitosan (B) = ZnONPs, 1% (0.02%) wt% chitosan (C) = ZnONPs, 2% (0.04%) wt% chitosan (D) = ZnONPs, 3% (0.06%) wt% chitosan (E) = 1% ZnONPs/1% PPE (0.02%) wt% of chitosan (F) = ZnONPs 2%/PPE2% (0.04%) wt% of chitosan (G) = ZnONPs 3%/PPE3% (0.06%). wt% of chitosan.

**Table 10 materials-16-01569-t010:** Effect of edible film and coatings on anthocyanin% of manfalouty pomegranate during storage at 2 °C.

Treatments	A (PPE), 5% (0.1%)	ZnONPs			ZnONPs/PPE		
		B 1% (0.02%)	C 2% (0.04%)	D 3% (0.06%)	E 1% (0.02%)	F 2% (0.04%)	G 3% (0.06%)
Zero time	18.95	18.95	18.95	18.95	18.85	18.85	18.85
5	17.55	18.25	17.88	17.38	18.12	17.66	17.10
10	16.42	17.86	16.90	16.48	17.52	16.52	16.22
15	-	17.23	16.18	15.80	17.10	15.80	15.38
20	-	16.56	15.46	14.73	16.34	15.22	14.62
L.S.D		S = 1.17		T = 1.49		S&T = 0.19	

(-) a spoiled reject samples. LSD Treatments = T LSD Storage period = S LSD (Storage period ∗ Treatments) = T ∗ S. Means within a column showing the same letters are not significantly different (*p ≥* 0.05). (A) = PPE extracts, 5% (0.1%) wt% chitosan (B) = ZnONPs, 1% (0.02%) wt% chitosan (C) = ZnONPs, 2% (0.04%) wt% chitosan (D) = ZnONPs, 3% (0.06%) wt% chitosan (E) = 1% ZnONPs/1% PPE (0.02%) wt% of chitosan (F) = ZnONPs 2%/PPE2% (0.04%) wt% of chitosan (G) = ZnONPs 3%/PPE3% (0.06%). wt% of chitosan.

**Table 11 materials-16-01569-t011:** Effect of edible film and coatings on total microbial count of manfalouty pomegranate during storage at 2 °C.

Treatments	A (PPE), 5% (0.1%)	ZnONPs			ZnONPs/PPE		
		B 1% (0.02%)	C 2% (0.04%)	D 3% (0.06%)	E 1% (0.02%)	F 2% (0.04%)	G 3% (0.06%)
Zero time	0.37	0.37	0.37	0.37	0.37	0.37	0.37
5	2.67	1.70	1.95	1.99	1.82	2.10	2.12
10	3.10	2.25	2.60	2.78	2.34	2.93	2.94
15	-	2.86	3.16	3.55	2.92	3.49	3.75
20	-	3.82	3.70	3.36	3.83	3.66	3.12
L.S.D		S = 1.14		T = 1.47		S&T = 0.16	

(-) a spoiled reject samples. LSD Treatments = T LSD Storage period = S LSD (Storage period ∗ Treatments) = T ∗ S. Means within a column showing the same letters are not significantly different (*p* ≥ 0.05). (A) = PPE extracts, 5% (0.1%) wt% chitosan (B) = ZnONPs, 1% (0.02%) wt% chitosan (C) = ZnONPs, 2% (0.04%) wt% chitosan (D) = ZnONPs, 3% (0.06%) wt% chitosan (E) = 1% ZnONPs/1% PPE (0.02%) wt% of chitosan (F) = ZnONPs 2%/PPE2% (0.04%) wt% of chitosan (G) = ZnONPs 3%/PPE3% (0.06%). wt% of chitosan.

**Table 12 materials-16-01569-t012:** Effect of edible film and coatings on moulds and yeast of manfalouty pomegranate during storage at 2 °C.

Treatments	A (PPE), 5% (0.1%)	ZnONPs			ZnONPs/PPE		
		B 1% (0.02%)	C 2% (0.04%)	D 3% (0.06%)	E 1% (0.02%)	F 2% (0.04%)	G 3% (0.06%)
Zero time	0.16	0.16	0.16	0.16	0.16	0.16	0.16
5	1.65	1.15	1.35	1.57	1.18	1.45	1.75
10	1.86	1.43	1.62	1.86	1.52	1.79	1.98
15	-	1.66	1.84	2.15	1.73	1.98	2.46
20	-	1.89	2.38	2.75	1.46	2.17	2.46
L.S.D		S = 1.15		T = 1.54		S&T = 0.12	

(-) a spoiled reject samples. LSD Treatments = T LSD Storage period = S LSD (Storage period ∗ Treatments) = T ∗ S. Means within a column showing the same letters are not significantly different (*p* ≥ 0.05). (A) = PPE extracts, 5% (0.1%) wt% chitosan (B) = ZnONPs, 1% (0.02%) wt% chitosan (C) = ZnONPs, 2% (0.04%) wt% chitosan (D) = ZnONPs, 3% (0.06%) wt% chitosan (E) = 1% ZnONPs/1% PPE (0.02%) wt% of chitosan (F) = ZnONPs 2%/PPE2% (0.04%) wt% of chitosan (G) = ZnONPs 3%/PPE3% (0.06%). wt% of chitosan.

**Table 13 materials-16-01569-t013:** Effect of edible film and coatings on color [hue angle (*h◦*)] t of manfalouty pomegranate during storage at 2 °C.

Treatments	A (PPE), 5% (0.1%)	ZnONPs			ZnONPs/PPE		
		B 1% (0.02%)	C 2% (0.04%)	D 3% (0.06%)	E 1% (0.02%)	F 2% (0.04%)	G 3% (0.06%)
Zero time	88.06	86.09	87.47	87.80	86.42	87.44	88.20
5	86.60	85.45	86.30	86.57	85.10	86.42	86.75
10	78.23	81.41	82.32	83.46	82.32	83.49	84.28
15	-	79.21	78.80	79.40	1579.42	80.48	80.86
20	-	74.49	75.19	76.40	75.76	76.45	77.46
L.S.D		S = 1.13		T = 1.58		S&T = 0.17	

(-) a spoiled reject samples. LSD Treatments = T LSD Storage period = S LSD (Storage period ∗ Treatments) = T ∗ S. Means within a column showing the same letters are not significantly different (*p ≥* 0.05). (A) = PPE extracts, 5% (0.1%) wt% chitosan (B) = ZnONPs, 1% (0.02%) wt% chitosan (C) = ZnONPs, 2% (0.04%) wt% chitosan (D) = ZnONPs, 3% (0.06%) wt% chitosan (E) = 1% ZnONPs/1% PPE (0.02%) wt% of chitosan (F) = ZnONPs 2%/PPE2% (0.04%) wt% of chitosan (G) = ZnONPs 3%/PPE3% (0.06%). wt% of chitosan.

## Data Availability

The results and data were analyzed in the Institute of Food Technology, Department of Food Processing and Packaging Engineering.

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
