# Peer review of "Nano Edible Coatings and Films Combined with Zinc Oxide and Pomegranate Peel Active Phenol Compounds Has Been to Extend the Shelf Life of Minimally Processed Pomegranates"

_materials, 2023, doi:10.3390/ma16041569_

Round 1

Reviewer 1 Report (Previous Reviewer 1)

Manuscript review "Influence of nano edible coating & films combined with zinc oxide and active phenol compounds from pomegranate peel to prolong shelf life of minimally processed pomegranate" written by Hosam Aboul Anean, L.O. Mallasi, Dina.M.D.Bader and Heba. A. shaat.
General remark to the work - negligence in the design! This is very disturbing! Sometimes it is very difficult to understand what is plotted along the coordinate axes, sometimes it is impossible to make out the details on the graphs, sometimes I cannot understand what is written ... It is impossible to publish the manuscript in this form, you need to painstakingly correct it!
It is not clear why the concentrations of ZnO and Pomegranate Peel Extracts presented in the manuscript were chosen. What is the reason for this choice? How does this correlate with other similar studies?
The results are not well presented.
A0%
B0.02%
C 0.04%
D0.06%
E 0.02%
F0.04%
G 0.06%
Constantly need to look at the table what it means. The authors need to correct this situation.
The data on viscometry are described extremely superficially. A deeper analysis of the results obtained is needed. At least you should write if there is a significant difference in different groups! Why are different groups represented in different graphs?
Measurements of hydrodynamic radius and zetta potential are too ambiguous. Too large Poly Dispersity indexes. What can these results indicate?
The microscopy data are interesting, but it is difficult to draw unambiguous conclusions from the description.
Most of the data is not statistically processed and does not even have standard errors of the mean. It is very difficult to understand whether there are differences from the control or not!
Line 199. "2- Determination of nano particles Zeta size : Device Type Malvern Made in UK and 195 Model: Zeta sizer nano series (Nano ZS). Size range (nm): 0.6: 6000 nm Size and zeta range 196 (mV) : (-200: 200mV)." The authors should give a more intelligible description of the methodology.

Author Response

Dear doctor

We inform you that all the question in the research have been answer

Zinc oxide - Zeta potential- Statistical analysis - Viscosity -Setting the language -Zinc toxicity -SEM -Rework the Research-Structural with thickness

Reviewer 2 Report (New Reviewer)

Dear Authors,

There are a lot of typos and errors in the text, both linguistic and editorial. English is poor, the text is sloppy.

You suggested at one point that the text is both review and experimental work. For the sake of clarity and ease of use by future readers, I suggest to divide this material into two works - a scientific review and experimental one.

In lines 72-72 it was suggested that ZnONPs-based coatings cause toxicity - this is quite an important aspect due to the subject of the work, it is necessary to develop this thread.

Lines 324-325: no short introduction to the results and discussion section.

Charts are wavyly edited. Axle markings and descriptions are in the wrong places. Besides, if you compare charts, they should have the same scale on the axes. Some of the graphs could be otherwise edited (e.g. zeta potential).

In the case of SEM, it is necessary to present pictures with the same magnification! Showing picures side by side with magnification of 140 and 2400 is non-intuitive and may mislead the reader.

There are no units in the tables.

I regret to say that the work may be interesting, but unfortunately it is not suitable for publication. Due to the poor quality of the text and the presented graphs and pictures, the work should be thoroughly redrafted.

Author Response

Zinc oxide-Zeta potential -Statistical analysis -Viscosity -Setting the language- Viscosity-Setting the language -Zinc toxicity -SED- Rework the search -Structural with thickness

Reviewer 3 Report (New Reviewer)

The article is devoted to the study of the effect of applying chitosan in the form of a film coating on zinc oxide nanoparticles, as well as changes in their properties, such as light transmission, strength and maximum elongation, film solubility, etc. In general, the presented work relates to the section of green chemistry and the production of nanostructures using green chemistry methods. The presented article has novelty and practical significance, and can also be recommended for publication in this journal, taking into account the comments of the reviewer, to which the authors should respond before accepting it for publication.

1 The abstract describes in sufficient detail the types of samples that were obtained, the authors should simplify the description, and also write in more detail about the results obtained and the effect of the concentration of the components on the storage stability of materials, etc.

2 The quality of the figures presented in the article requires significant revision, firstly, it is necessary to present better scales, and also to exclude the imposition of labels on the axes. Measurement errors should be given in order to determine the dynamics of changes in the obtained dependencies, since with the introduction of measurement errors, the describing curves can be presented in a different form, and the interpretation of the data will be presented more clearly.

3 The same applies to all the data presented in the table, for all measured values, the error range should be given, since a number of values ​​have small changes that can be leveled by the measurement error and the absence of significant changes.

4 Morphological data presented using electron microscopy require processing, more detailed images should be provided, reflecting morphological features.

5 The authors talk about the formation of fairly complex composite structures, but do not provide data on the isotropy of particle distribution in films or isotropy over thickness.

Author Response

We inform you that all question the research have been answered

Zinc oxide-Zeta potential -Statistical analysis Viscosity -Setting the language - Zinc toxicity - SEM -Rework the search -Structural with thickness

Round 2

Reviewer 1 Report (Previous Reviewer 1)

The authors answered most of my questions and improved the manuscript significantly.

Reviewer 3 Report (New Reviewer)

The authors answered all the questions, the article can be accepted for publication.

This manuscript is a resubmission of an earlier submission. The following is a list of the peer review reports and author responses from that submission.

Round 1

Reviewer 1 Report

Manuscript review "Influence of nano edible coating & films combined with zinc oxide and active phenol compounds from pomegranate peel to prolong shelf life of minimally processed pomegranate" written by Hosam Aboul Anean, L.O. Mallasi, Dina.M.D.Bader and Heba. A. shaat.
General remark to the work - negligence in the design! This is very disturbing! Sometimes it is very difficult to understand what is plotted along the coordinate axes, sometimes it is impossible to make out the details on the graphs, sometimes I cannot understand what is written ... It is impossible to publish the manuscript in this form, you need to painstakingly correct it!
It is not clear why the concentrations of ZnO and Pomegranate Peel Extracts presented in the manuscript were chosen. What is the reason for this choice? How does this correlate with other similar studies?
The results are not well presented.
A0%
B0.02%
C 0.04%
D0.06%
E 0.02%
F0.04%
G 0.06%
Constantly need to look at the table what it means. The authors need to correct this situation.
The data on viscometry are described extremely superficially. A deeper analysis of the results obtained is needed. At least you should write if there is a significant difference in different groups! Why are different groups represented in different graphs?
Measurements of hydrodynamic radius and zetta potential are too ambiguous. Too large Poly Dispersity indexes. What can these results indicate?
Figures 5 and 6 must be replaced!
The microscopy data are interesting, but it is difficult to draw unambiguous conclusions from the description.
Most of the data is not statistically processed and does not even have standard errors of the mean. It is very difficult to understand whether there are differences from the control or not!
The introduction should be divided into 3 paragraphs. The first one is about chitosan. The second is about ZnO nanoparticles. The third is about pomegranate fruits.
Line 71. "Zinc oxide (ZnO) is an eco-friendly, non-toxic to human cells with noble properties.." Zinc is a highly toxic chemical element. Perhaps the authors were wanting to write that zinc oxide has a more pronounced effect on microorganisms on eukaryotic cells, as shown in the work (https://doi.org/10.3390/ma14216586). ???
Line 116. "Manfalouty' pomegranate (Punica granatum L.) fruits..." The Latin should be written in italics.
Line 195. "2- Determination of nano particles Zeta size : Device Type Malvern Made in UK and 195 Model: Zeta sizer nano series (Nano ZS). Size range (nm): 0.6: 6000 nm Size and zeta range 196 (mV) : (-200: 200mV)." The authors should give a more intelligible description of the methodology.

Author Response

The research and statistical analysis was done

Reviewer 2 Report

Following are my comments regarding the present manuscript titled ‘Influence of nano edible coating & films combined with zinc oxide and active phenol compounds from pomegranate peel to prolong shelf life of minimally processed pomegranate’.

There are serious issues with respect to comprehensibility of the manuscript. The manuscript is incoherent, both in terms of language and the technical aspects (abstract, objective, methods, results). Specific comments are below:

1. Abstract is incomprehensible. Sample statements from abstract: “The effect of edible coating and film from chitosan and the incorporating it with action of 15 ZnONPs with active phenol compounds from extracts pomegranate peel(PPE) and the rheological 16 properties, partical size distribution, zeta potentioal, color, Light transmittance and scanning elec-17 tron microscopy of the prepared films were determined.” “The thickness, tensile strength, elongation, 18 %solubility of produced by edible film with zinc oxide ZnONPs with active phenol compounds 19 from extracts pomegranate peel(PPE)”.

2. Objective of the study is unclear: The manuscript has two different statements – “In this sense, the main goal of this article is to review and update the information available 107 on the use of edible coatings as carriers of food ingredients antimicrobials to improve the 108 safety quality and functionality of fresh fruits. (Pranoto et al., 2005).” And “The objective of this work is ZnONPs and active phenol compounds from extracts pomegranate peel(PPE) na-110 noparticles loaded on chitosan films , and physical & mechanical, viscosity , vapor , zeta 111 particle size emulsion, color, Light transmittance and SEM to select the best edible coating 112 previous check to improve pomegranate fruit storability under cold storage.” So finally what is the objective?

3. Other statements from body of manuscript:

“The study prepared of rheological property such as shear rate, shear stress and vis-327 cosity of samples were measured prepared nano edible film and coatings at different treat-328 ment’s A 0%, B 0.02%, C 0.04%, D 0.06%, E 0.02%, F 0.04% and G 0.06% and different 329 shear rates ( 13.2, 26.4, 39.6 ,52.8 ,66.00, 79.2 1/s). Figure, (1, 2 ,3 and 4) and table (2).”

“The results obtained are shown in table (2) & fig (5), at treatments (E20% , F40% and 421 G60%) was higher than the concentration of the treatments A 0%, B 0.02%, C 0.04%, D 422 0.06%, E 0.02%, F 0.04% and G 0.06% in both poly disparity index (PdI) and hydrody-423 namic diameter of particle size(nm), it was found that the peak was (0.648 , 0.503 and 0.771 424 and 134, 824.6 and 966 nm) for B 0.04%, C 0.08% and D 0.12% respectively.”

“The effect of edible coating as have a high potential to carry active ingredients 827 such as nano materials maintained weight loss, total soluble solids, total acidity, ascor-828 bic acid, total count and mould & yeast quality attributes during storage at 2ºC of pom-829 egranate and relative humidity 90-95% .”

 4. It appears like there are lines of extra text in the manuscript, which might have been added by the authors themselves, or by a reviewer. Samples below:

“Explain these microscopic images seven prepared separate cellular nano biopolymer 499 production for batch fermentation of bacterial to produce edible film and after taking 500 the cross-section the morphology of surface was investigated using SEM images (Fig.7), 501 and the surface roughness was estimated using nano edible coating & films.”

“Explain the relation between shear rate, apparent viscosity and 334 shear stress for different samples.”

Therefore, the authors have to go through the text and rectify all the issues regarding objective, abstract, methods, results, language and other aspects for enabling proper review.

Author Response

(The authors gave the same response as above.)

Round 2

Reviewer 1 Report

The authors forgot to answer the reviewer's questions. Please do it.

Author Response

dear reviewers

resarch paper has been modified after second round of revision

Reviewer 2 Report

The authors do not seem to have addressed the earlier comments. They are requested to do so. 

Manuscript is still incomprehensible. Abstract, objective, methods, results are unclear.

The additional text in the current version (highlighted in yellow) does not add value in addressing the underlying issues with the manuscript.

Author Response

Dear reviewers

research paper has been modified after second round of revision

kind regards
